# Effect of Selenium and Lycopene on Radiation Sensitivity in Prostate Cancer Patients Relative to Controls

**DOI:** 10.3390/cancers15030979

**Published:** 2023-02-03

**Authors:** Varinderpal S. Dhillon, Permal Deo, Michael Fenech

**Affiliations:** 1Health and Biomedical Innovation, Clinical and Health Sciences, University of South Australia, Adelaide 5000, Australia; 2Genome Health Foundation, North Brighton 5048, Australia

**Keywords:** micronutrients, micronuclei, radiation sensitivity, prostate cancer (PC), DNA damage biomarkers

## Abstract

**Simple Summary:**

Prostate cancer (PC) is the most common cancer in elderly men, but its prevention by appropriate dietary interventions remains elusive. In this study, we show that the white blood cells of PC patients presented more DNA damage than those of healthy controls and were more prone to DNA damage induced by ionising radiation. Furthermore, our results indicate that this excess of DNA damage can be explained by low levels of lycopene (a carotenoid found in some red fruits such as tomatoes) and selenium (a mineral found in protein-rich foods such as beef and Brazil nuts). The results of this study suggest that a higher intake of foods rich in lycopene and selenium may help reduce the risk of prostate cancer and DNA damage caused by ionising radiation and/or oxidative stress.

**Abstract:**

Almost half of prostate cancer (PC) patients receive radiation therapy as primary curative treatment. In spite of advances in our understanding of both nutrition and the genomics of prostate cancer, studies on the effects of nutrients on the radiation sensitivity of PC patients are lacking. We tested the hypothesis that low plasma levels of selenium and lycopene have detrimental effects on ionising radiation-induced DNA damage in prostate cancer patients relative to healthy individuals. The present study was performed in 106 PC patients and 132 age-matched controls. We found that the radiation-induced micronucleus (MN) and nuclear buds (NBuds) frequencies were significantly higher in PC patients with low selenium (*p* = 0.008 and *p* = 0.0006 respectively) or low lycopene (*p* = 0.007 and *p* = 0.0006 respectively) levels compared to the controls. The frequency of NBuds was significantly higher (*p* < 0.0001) in PC patients who had low levels of both selenium and lycopene compared to (i) controls with low levels of both selenium and lycopene and (ii) PC patients with high levels of both selenium and lycopene (*p* = 0.0001). Our results support the hypothesis that low selenium and lycopene levels increase the sensitivity to radiation-induced DNA damage and suggest that nutrition-based treatment strategies are important to minimise the DNA-damaging effects in PC patients receiving radiotherapy.

## 1. Introduction

Prostate cancer is one of the most common non-cutaneous cancers and the fifth leading cause of cancer deaths among men worldwide [1]. It is worth noting that the burden of the disease will soar due to aging and the economic boom [2]. Factors such as ethnic background, family history and advancing age are associated with an increased risk of this disease [3,4]. Therefore, it is important to identify high-risk individuals for the precise management of the disease and develop better cancer biomarkers for timely preventative interventions.

Selenium is an essential mineral and has antioxidant properties. Increased oxidative stress due to the generation of reactive oxygen species (ROS) likely increases the adverse effects after radio- and chemotherapeutic management of the disease [5]. Dietary selenium supplementation may protect the healthy tissues and reduce the side effects of genotoxic treatments. In spite of its antioxidant and anti-carcinogenic properties, the results for selenium are inconsistent and conflicting [6,7,8,9,10]. Recently, data from 15 prospective studies showed some encouraging results that provided evidence of an association of high blood selenium levels and with a reduced risk of aggressive prostate cancer [11].

Like selenium, lycopene also plays an important protective role in the prevention of cancer, including prostate cancer, by inhibiting oxidative stress, apoptosis and inflammation due to its anti-inflammatory, anti-oxidative and anti-proliferative properties [12,13,14]. The lycopene levels in the blood are inversely associated with the risks of cardiovascular disease, metabolic syndrome and cancer, including prostate cancer [15,16]. A double-blind placebo-controlled study involving lycopene-rich juices showed a significant increase in the serum lycopene levels, leading to a reduction in genome damage, the generation of ROS and lessen disease burden [17]. It has been shown that the administration of lycopene to newly diagnosed PC patients for three weeks, twice a week, resulted in lowering the disease risk and the growth of PC cells [18]. It has also been shown that men with localised prostate adenocarcinoma, receiving tomato sauce (lycopene, 30 mg/day for weeks), showed (i) an increase in lycopene levels in the serum and prostate tissue, (ii) a reduction in PSA concentration and (iii) lower DNA damage in the prostate gland and white blood cells [19].

The precision therapeutic treatment of patients undergoing radiotherapy remains an aim of clinical testing to predict better outcomes prior to treatment. The impact of genetics, epigenetics and life style factors including diet needs to be better understood to achieve precise and desired outcomes after a therapeutic treatment [20]. In spite of the molecular mechanisms involved in radio-biological processes being well understood, the challenge still remains of how to better identify individuals at increased adverse therapeutic risk [21]. Exposure to radiations during cancer radiotherapy results in increased toxicity, oxidative stress and inflammation that can lead to genetic instability [22]. Chromosome aberrations such as deletions and rearrangements are the main cancer-initiating events. Therefore, it is important to identify people at increased risk of developing cancer by using biomarkers of chromosome damage, such as micronuclei (MNi), nucleoplasmic bridges (NPBs) and nuclear buds (NBUDs) [23,24,25]. MNi are small extra-nuclear bodies that contain damaged chromosome fragments and/or whole chromosomes that were not incorporated into the main nucleus after cell division [24]. MNi are a cancer-predictive biomarkers of chromosome breakage and/or of the loss of whole chromosomes. NPBs are biomarkers of dicentric chromosomes caused by DNA misrepair and/or telomere end-fusions, and NBUDs are biomarkers of the elimination of amplified DNA and/or unresolved DNA repair complexes [24]. A higher MN frequency reflects a higher genomic damage and may thus be used as a marker for predicting the cancer risk [25]. The cytokinesis-block micronucleus (CBMN) assay is one of the most important in vivo and in vitro cytogenetic assays [24,26]. It is a comprehensively validated and standardised method to evaluate in vivo the radiation exposure of individuals working in healthcare (occupational exposure) and of accidentally exposed individuals [27,28]. Keeping in view the antioxidant potential of both selenium and lycopene, we tested the hypothesis that the peripheral blood lymphocytes of PC patients have an aberrantly increased genomic instability and are susceptible to ionizing radiation-induced (3 Gy) DNA damage due to low levels of selenium and lycopene.

## 2. Materials and Methods

### 2.1. Study Population

The prostate cancer project is a collaborative initiative undertaken by researchers from Royal Adelaide Hospital and CSIRO. The Human Ethics Committees of Royal Adelaide Hospital and CSIRO approved the study design. All subjects including controls provided written informed consent to participate in the project. All prostate cancer patients (n = 103) at the time of recruitment were untreated, and the diagnosis was confirmed by histopathological findings that included suspicious digital rectal examination (DRE) and significantly higher prostate-specific antigen (PSA; 0.08–45 ng/mL) levels in the blood. The Gleason score [29] varied between 6 and 9 at the time of the diagnosis. Age-matched controls (n = 132) who were healthy and prostate cancer-free (normal PSA levels ranging between 0.0 and 3.0 ng/mL) at the time of recruitment and were not taking any medication for life-threatening diseases were included in the study. We intended to match the cases and controls with regard to their smoking status; however, it was not possible, due to the difficulty in recruiting smokers in the control group.

### 2.2. Blood Collection and Irradiation of Lymphocytes

Blood samples were collected after overnight fasting in lithium heparin tubes from PC patients and controls. An hour after blood collection, 500 μL of whole blood was mixed with 4.5 mL of pre-warmed RPMI-1640 culture medium (Thermo Trace, Melbourne, Australia) supplemented with 10% foetal calf serum (FCS; Thermo Trace, Melbourne Australia). To cause radiation-induced DNA damage in lymphocytes, the whole blood cultures were exposed to 3 Gy γ-rays from a 137Cs source (Cis Bio IBL 437 C Blood Product Irradiator, dose rate 5.34 Gy/min). The irradiation dose selected in the present study is known to induce approximately a 100-fold increase in the micronucleated (MN) binucleated (BN) frequency relative to spontaneous frequency in cultured lymphocyte using CBMN cyt assay [28].

### 2.3. Cytokinesis-Block Micronucleus Cytome (CBMN Cyt) Assay for Lymphocytes Using Whole Blood Cultures

The assay was performed as described previously [24] with slight modifications. The whole blood cultures were set up in duplicate. For comparing the results, unirradiated blood cultures were used as controls. Following radiation exposure (radiation-induced damage), irradiated and unirradiated cultures were incubated for 1 h in a humidified incubator at 37 °C containing 5% CO_2_. Following this incubation, 45 μL of phytohaemagglutinin (PHA, 22.5 mg/mL; Jomar Diagnostics, Stepney, Australia) was added to each culture, and the cultures were incubated for further 44 h prior to the addition of cytochalasin-B (Cyto-B; Sigma, Macquarie Park, Australia) to a final concentration of 6 μg/mL. Following the addition of Cyto-B, the cultures were incubated for another 24 h. The lymphocytes were separated from these cultures by carefully overlaying the evenly distributed culture contents with 1.5 mL Ficoll-Paque (Amersham BioSciences, Buckinghamshire, UK ) in TV10 tubes (Sarstedt, Mawson Lakes, Australia). The tubes were then centrifuged for 30 min at 400× *g* at 20 °C. The isolated buffy lymphocyte layer (~200 μL) was transferred to an another TV10 tube containing 600 μL of Hanks’ balanced salt solution (HBSS; Thermo Trace, Melbourne, Australia) and centrifugated at 180× *g* at 20 °C for 10 min. The supernatant was discarded, and the cells (lymphocytes) were re-suspended in 300 μL of RPMI-1640 culture medium containing 5.0 μL of dimethyl sulfoxide (DMSO; Sigma, Macquarie Park, Australia) to facilitate the disaggregation of the cells. The cells were then transferred onto slides using a cytocentrifuge (Shandon, Runcorn, UK). The air-dried slides were fixed and stained using Diff-Quik (LabAids, Narrabeen, Australia). The slides were scored under code for BN cells containing MN, NPB and NBuds, as per previously described scoring criteria [24]. At least 1000 BN cells were scored per slide that consists of MN, NPB and NBuds. Representative examples of these biomarkers are depicted in Figure 1.

### 2.4. Micronutrient Analysis and PSA Levels

The plasma concentrations of lycopene and selenium were measured using HPLC at the CSIRO Human Nutrition laboratory [30]. Duplicate analyses (homogeneity of the sample) were carried out for each sample. The PSA levels in blood were measured by a certified medical diagnostics laboratory using an immunoassay in the healthy controls and PC cases.

### 2.5. Statistical Analysis

All data for these two nutrients and other parameters were analysed for Gaussian distribution to determine whether to use parametric or non-parametric tests. We used the χ^2^-square test if there were significant differences in the smoking status in both PC patients and controls. The unpaired non-parametric Student’s *t* test was used to determine the significance of the differences between the two groups with regard to age, PSA levels, baseline and radiation-induced micronuclei, nucleoplasmic bridges and nuclear buds in bi-nucleated cells. The results with respect to the DNA damage biomarkers were analysed using one-way ANOVA in relation to high or low selenium and high or low lycopene concentrations (high blood concentrations were >120 μmol and >0.25 μg/mL, and low blood concentrations were ≤120 μmol and ≤0.25 μg/mL for selenium and lycopene, respectively). These cut-off values were based on the median concentrations in healthy controls. All analyses were performed using PRISM 9.0 (GraphPad software, San Diego, CA, USA). All *p* values < 0.05 were considered statistically significant.

## 3. Results

### 3.1. Demographic and Clinical Characteristics of the Cohorts

Table 1 shows the demographic profiles of the PC patients as well the as the healthy controls. There were no significant differences in the age of the cases (mean age 71.24 ± 7.18 years) and the controls (mean age 69.07 ± 7.99 years). The number of smokers was significantly higher (current and ex-smokers; *p* = 0.0001) in the patient group compared to the controls. Similarly, the PSA levels were significantly higher (*p* = 0.0001) in the PC cases compared to the controls.

### 3.2. Association of Low Selenium Concentrations with DNA Damage Biomarkers at Baseline and after a 3 Gy Radiation Challenge in Controls and Patients

The results indicated that the baseline MNi frequency was marginally higher in PC patients with low or high selenium levels when compared to controls (Figure 2A), but the difference was not statistically significant. Similar results were obtained for both groups when the plasma selenium concentration was high. When the lymphocytes from the controls and PC patients with low plasma selenium levels were irradiated with 3 Gy, the MN frequency was significantly higher in the patients than in the controls (*p* = 0.008; Figure 2B), whereas it was slightly higher (*p* = 0.7; Figure 2B) in the PC cases compared to the controls in the presence of high plasma selenium levels.

The baseline frequency of NPBs was significantly higher in the PC cases with low selenium than in the controls (*p* = 0.005; Figure 2C). Similarly, in the PC cases with high selenium levels, the baseline frequency of NPBs was marginally significantly higher compared to the controls (*p* = 0.05; Figure 2C). The radiation-induced frequency of NPBs also showed a similar trend but did not reach significant level in both the PC cases and the controls, with either low or high selenium levels (Figure 2D). The baseline frequency of NBuds was not significantly different in the PC cases and controls with either low or high selenium levels, though it was marginally higher in the PC group (Figure 2E). However, the radiation-induced frequency of NBuds was significantly higher in the PC cases compared to the controls with either low or high selenium levels (*p* = 0.0006 and *p* = 0.055 respectively; Figure 2F).

### 3.3. Association of Low Lycopene Concentration with DNA Damage Biomarkers at Baseline and after a 3 Gy Radiation Challenge in Controls and Patients

The baseline MN frequency was not significantly different in the PC cases and controls with either low or high lycopene levels (Figure 3A), though it was marginally higher in the PC cases. However, the radiation-induced MN frequency was significantly higher in the PC cases than in the controls with low lycopene levels (*p* = 0.007; Figure 3B), whereas it was marginally higher in the PC cases with high lycopene concentration.

The baseline frequency of NPBs was significantly higher in the PC cases with either low or high lycopene levels compared to the controls (*p* = 0.002 and *p* = 0.01, respectively; Figure 3C). The radiation-induced frequency of NPBs as marginally higher in the PC cases compared to the controls, irrespective of the lycopene status (Figure 3D). The baseline frequency of NBuds was marginally higher in the PC cases compared to the controls, irrespective of the lycopene concentration (Figure 3E). The radiation-induced frequency of NBuds was significantly higher in the PC cases compared to the controls with either low or high lycopene status (*p* = 0.0006 and *p* = 0.05, respectively; Figure 3F). Generally, a low lycopene concentration was associated with increased DNA damage biomarkers, especially in prostate cancer cases.

### 3.4. Cumulative Effects of Low Selenium and Lycopene on DNA Damage Biomarkers at Baseline and after a 3 Gy Radiation Challenge in Controls and Patients

The baseline MN frequency was significantly higher in both controls (*p* = 0.007) and PC patients (*p* = 0.0002) with low selenium and lycopene levels compared to subjects with high selenium and lycopene concentrations (Figure 4A). Similarly, the radiation-induced MN frequency was significantly higher in the controls (*p* = 0.0003) and PC cases (*p* = 0.007) with low selenium and lycopene levels compared to subjects with high selenium and lycopene levels (Figure 4B). The baseline and radiation-induced MN frequencies were elevated in the PC cases compared to the controls, irrespective of the selenium and lycopene concentrations. Similar results were observed with regard to the baseline and radiation-induced frequencies of NPBs in the PC cases and controls, irrespective of the selenium and lycopene concentrations (Figure 4C,D). The baseline frequency of NBuds was not significantly different in the PC cases and controls, irrespective of the selenium and lycopene status; however, it was slightly higher in the PC cases (Figure 4E). The radiation-induced frequency of NBuds was significantly higher in the PC cases compared to the controls when the selenium and lycopene concentrations were low (*p* < 0.0001; Figure 4F); however, it did not reach a significant level when the selenium and lycopene levels were high. The radiation-induced frequency of NBuds was significantly higher in the controls (*p* < 0.0001) and PC cases (*p* < 0.0001) with low selenium and lycopene levels compared to subjects with high selenium and lycopene concentrations (Figure 4F). The results of samples with high/low or low/high selenium and lycopene levels were not significantly different (Appendix A).

Our results indicate that a low selenium and lycopene status plays an important role in aggravating the DNA damage in response to radiation, and its impact is significantly pronounced in PC cases compared to healthy controls.

## 4. Discussion

Cancer is viewed as a progressively multistep process that involved the mutation and selection of cells with increasing capacity for proliferation, invasion and accumulation of genomic damage; there is evidence that cancer has a genetic basis [31,32]. PC is the most common non-cutaneous solid malignancy in men worldwide. PC patients take supplements and natural products including selenium and lycopene to improve cancer outcomes as a chemo-preventive strategy; however, the evidence of a benefit from these nutritional interventions so far is very limited; therefore, further research is required to fully understand the role of these nutrients in improving cancer outcomes [33,34,35].

The CBMN cytome assay endpoints provide information about chromosome breakage and rearrangements as well as gene amplification in cultured peripheral blood lymphocytes [24,26,36,37]. The lymphocytes circulate in the human body, can accumulate genetic damage and are exposed to a variety of tumour-derived substances as well as to altered tissue microenvironments. Therefore, acquired chromosome aberrations in these cells are important biomarkers of genomic instability that can predict an increased cancer risk [38,39,40,41]. In the present study, it was shown that cultured peripheral blood lymphocytes from PC patients with low selenium and lycopene blood levels exposed to 3 Gy radiation showed increased chromosome instability, as indicated by the increased frequencies of MN and NBuds compared to those measured in the controls. We reported previously that the frequency of radiation-induced NBuds was significantly higher in PC patients than in controls, whereas those of MN and NPBs were not significantly different in these groups [42].

It was also reported that the radiation-induced (3 Gy) MN frequency predicted increased gastrointestinal (GI) and genitourinary (GU) morbidity [43]. However, in our previous report, only the spontaneous MN frequency was found to be linked with the worsening of GI symptoms, significantly correlated with lower plasma concentrations of selenium and α-tocopherol [44]. These different findings were perhaps due to fact that the confounding effects of various nutritional factors including selenium and lycopene on the expression of the radiation-sensitivity phenotype (measured by the CBMN assay) were not examined in the previous study [43]. Therefore, based on the available data, it is plausible that patients who consume a diet deficient in selenium and lycopene may be more prone to a high radiation-induced DNA damage. The present results support the above hypothesis that people with low blood selenium and lycopene levels are sensitive to radiation-induced DNA damage and that this damage is significantly pronounced in PC patients. It has also been suggested that new dietary strategies should be diligently pursued to further understand the mechanisms associated with the progression of this disease [45].

Radiotherapy induces an arrest in the cell cycle in a p53-dependent manner in response to genetic damage induced by ionizing radiation, which can lead to the generation of free radicals that can attack the sugar-phosphate backbone of DNA [46,47]. Selenium is an integral part of extracellular and cellular metalloenzymes, glutathione peroxidase, thioredoxin reductase and other selenoproteins that have anti-inflammatory and/or antioxidant properties [48]. Recent in vitro and in vivo studies and findings from an umbrella review [48] have shown that lycopene has antioxidant [49], anti-inflammatory [50], anti-carcinogenic and cardio-protective properties [51], suggesting that it plays a protective role in chronic diseases including cancer [52]. Lycopene in association with selenium could lower oxidative stress, decrease lipid peroxidation, reduce the level of reactive oxygen species and scavenging free radicals generated following exposure to ionizing radiation, thereby reducing DNA damage. Inflammation is induced by exposure to radiations that can cause tissue damage due to the generation of ROS and to persistently high ROS levels as a consequence of damaged mitochondria and activated NADPH oxidases [53]. ROS are released from the mitochondria via the classic ATM–p53–bax DDR mechanism [54]. Many toll-like receptors (TLRs; important mediators of inflammatory pathways), RIG-1 (RNA sensing) and cGAS/cGMP/STING sensors connect the DNA damage response (DDR) to activate pro-inflammatory responses as a result of cellular stress by engaging NF-kB and TKB1/IRF3 pathways, thus promoting positive and negative feedback loops triggering senescence and cell death [55]. Radiation-induced DNA damage can lead to the activation of cytosolic DNA (extra-nuclear DNA present in micronuclei) sensing pathway mediated by the stimulation of interferon genes (STING) of the cGAS/cGMP/STING pro-inflammatory pathway.

To further understand how these nutrients, either alone or in combination, provide protection against radiation-induced DNA damage, it is important to understand their general metabolism that involves the following: (i) release from the food matrix, (ii) uptake by intestinal cells, (iii) secretion into the blood and circulation, (iv) tissue uptake and retention and (v) role of various proteins involved in these physiological processes. It is also important to understand the role of various genes including p53 [56], their variation in influencing the metabolic processes as wells as their precise role and mechanism(s) in eliciting the DNA damage response and efficient DNA repair.

## 5. Conclusions

In conclusion, the results of our study provide important evidence that prostate cancer patients who are deficient in selenium and/or lycopene may be more prone to DNA damage induced by ionising radiation. These observations provide a strong basis for future studies that test the feasibility of reducing DNA damage in PC patients by administering selenium and lycopene supplementation.

## Figures and Tables

**Figure 1 cancers-15-00979-f001:**
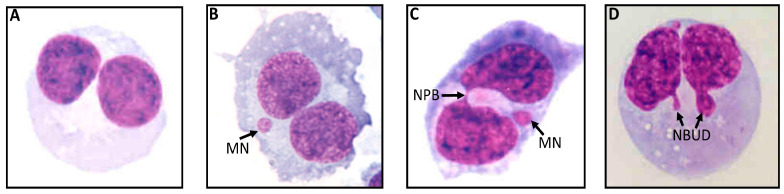
Representative examples of various CBMN cyt assay markers. (**A**) Binucleated (BN) cell without MN, NPB or NBuds, (**B**) BN cell with MN, (**C**) BN cell with NPB and (**D**) BN cell with NBuds.

**Figure 2 cancers-15-00979-f002:**
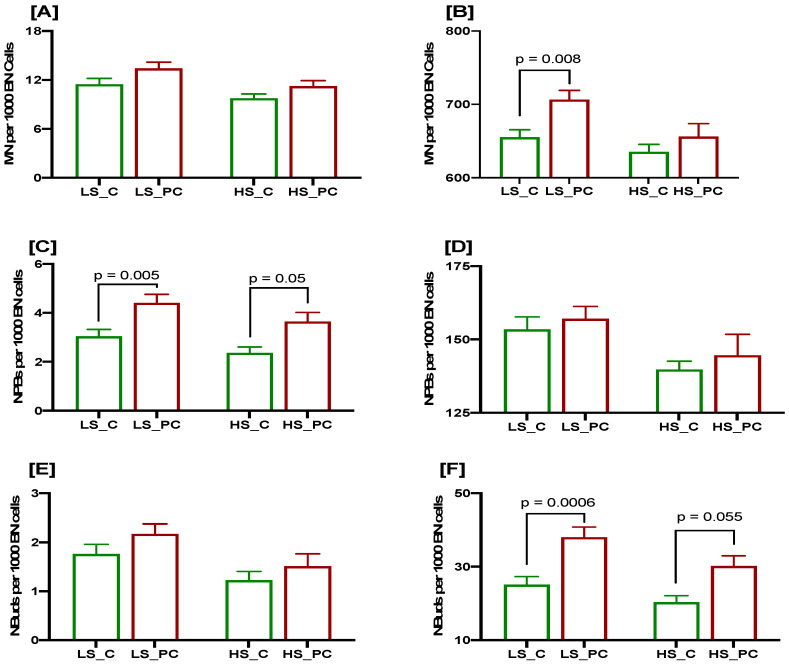
CBMN Cytome assay biomarkers in PC patients (red bars) and age-matched healthy controls (green bars) stratified as per selenium concentration (LS: low selenium; HS: High selenium); MN frequency at baseline (**A**) and after 3 Gy irradiation (**B**); NPBs at base line (**C**) and after 3 Gy irradiation (**D**); NBuds at baseline (**E**) and after 3 Gy irradiation (**F**). N = 63, 69, 69 and 34, respectively, from left to right for each bar. *p* values are only indicated for those comparisons that are significant.

**Figure 3 cancers-15-00979-f003:**
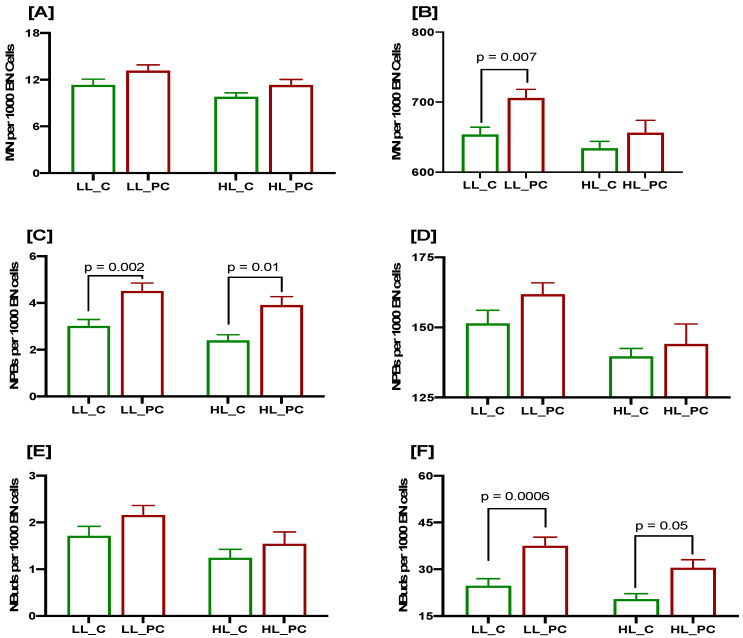
CBMN Cytome assay biomarkers in PC patients (red bars) and age-matched healthy controls (green bars) stratified as per lycopene concentration (LL: low lycopene; HL: high lycopene); MN frequency at baseline (**A**) and after 3 Gy irradiation (**B**); NPBs at baseline (**C**) and after 3 Gy irradiation (**D**); NBuds at baseline (**E**) and after 3 Gy irradiation (**F**). N = 61, 67, 71 and 36, respectively, from left to right for each bar. *p* values are only indicated for those comparisons that are significant.

**Figure 4 cancers-15-00979-f004:**
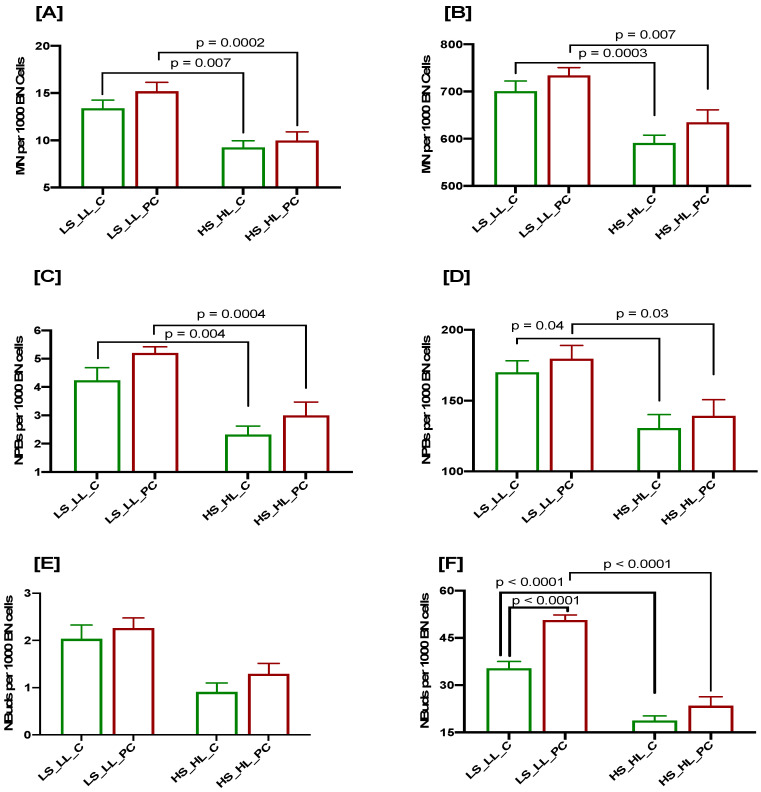
CBMN Cytome assay biomarkers in PC patients (red bars) and age-matched healthy control (green bars) stratified as per selenium and lycopene concentrations (LS: low selenium; HS: high selenium; LL: low lycopene; HL: high lycopene); MN frequency at baseline (**A**) and after 3 Gy irradiation (**B**); NPBs at baseline (**C**) and after 3 Gy irradiation (**D**); NBuds at baseline (**E**) and after 3 Gy irradiation (**F**). N = 29, 34, 52 and 28, respectively, from left to right for each bar. *p* values are only indicated for those comparisons that are significant.

**Table 1 cancers-15-00979-t001:** Comparison of prostate cases and controls by selected demographic, micronutrient and clinical variables.

Characteristics	Cases	Controls	*p* Value
Age (years; Mean ± SD)	71.24 ± 7.18	69.07 ± 7.99	0.88
Total plasma PSA (ng/mL; mean ± SD)	9.5 ± 8.5	2.4 ± 2.45	0.0001 *
Gleason score	6–9	-	
**Smoking status**			
Current smokers	9	3	0.0001 *
Ex-smokers	60	39	
Non-smokers	25	54	
Undeclared	24	36	
Selenium (µg/L)	116.1 ± 1.59 (71.83–157.60)	125.6 ± 2.56 (79.17 –238.10)	0.002 *
Lycopene (µg/L)	0.184 ± 0.011 (0.013–0.655)	0.215 ± 0.009 (0.03–0.58)	0.008 *

* Chi-square test. Values given in the brackets represent the range.

## Data Availability

Data will be uploaded to a publicly available repository upon acceptance of the manuscript.

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
