# Peer review of "Effect of Selenium and Lycopene on Radiation Sensitivity in Prostate Cancer Patients Relative to Controls"

_cancers, 2023, doi:10.3390/cancers15030979_

Round 1

Reviewer 1 Report

This is an interesting study aimed to investigate the association between blood selenium and lycopene concentrations on radiation induced DNA damage in lymphocytes from prostate cancer patients and control. In my opinion, the work is interesting, well conducted and discussed. Only few notes to improve the manuscript as follow:

1.     Some sentences are too long and not clear: examples are on lines 63-67 and 75-80.

2.     Provide an explanation for the abbreviation CBMN on line 87.

3.     Line 113: I think that Table 1 should be mentioned only in the results section.

4.     Lines 182-184: since the results are not statistically significant no differences exist between baseline MN frequency PC patients and controls (Fig 2A).

5.     In order to make clear the statistical analysis of the results (showed in figures 2, 3 and 4) carried out by ANOVA (which compare all groups), letters should be used on each column in which the means with the same letters are not statistically different and means without a common letter differ, p < 0.05.

6.     Why “unstratified” data are not shown?

Author Response

Reviewer 1

Comments and Suggestions for Authors

This is an interesting study aimed to investigate the association between blood selenium and lycopene concentrations on radiation induced DNA damage in lymphocytes from prostate cancer patients and control. In my opinion, the work is interesting, well conducted and discussed. Only few notes to improve the manuscript as follow:

  1. Some sentences are too long and not clear: examples are on lines 63-67 and 75-80.

Response: The two sentences on lines 63-67 and 75-80 were amended to make them briefer and/or clearer as indicated in the revised manuscript.

  1. Provide an explanation for the abbreviation CBMN on line 87.

Response: The explanation of CBMN assay is "cytokinesis-block micronucleus" assay. This explanation has been provided in the revised text.

  1. Line 113: I think that Table 1 should be mentioned only in the results section.

Response: as suggested we deleted the sentence "Table 1 describes…..controls" in lines 113, 114. Table 1 is now only mentioned in the result section

  1. Lines 182-184: since the results are not statistically significant no differences exist between baseline MN frequency PC patients and controls (Fig 2A).

Response: We amended the sentence in lines 182-184 to indicate the lack of statistical significance as follows: "The results indicate that baseline MNi frequency is marginally higher in PC patients with low or high selenium status when compared to controls (Fig 2A) but the difference was not statistically significant.

  1. In order to make clear the statistical analysis of the results (showed in figures 2, 3 and 4) carried out by ANOVA (which compare all groups), letters should be used on each column in which the means with the same letters are not statistically different and means without a common letter differ, p < 0.05.

Response: We already indicated the comparisons that are statistically significant by indicating the P values directly on the graphs. The use of letters to indicate significance of such differences is comparatively less clear and can be confusing. Therefore, we decided not to use the letters notation. However, we included the following statement in the legends of the relevant figures:

“P values are only indicated for those comparisons that are significant.”

  1. Why “unstratified” data are not shown?

Response: We did not show unstratified data because we wanted to emphasise the DNA damage differences between cases and controls and how these were affected depending on low or high selenium and/or low or high lycopene.

Reviewer 2 Report

This is my review on “Effect of Selenium and Lycopene on radiation sensitivity in Prostate Cancer patients relative to controls”. This study aims to elucidate the protective role of lycopene and selenium by food intake on prostate cancer. 

Introduction provides the necessary information and background of the study.   The study population and methodology are described in full details and the statistical analysis is the appropriate one. Results are interesting and presented clearly and well.  Discussion is supported by the results and provides new insight on the matter. 

Authors provide many perspectives for further study. 

Author Response

Reviewer 2

Comments and Suggestions for Authors

This is my review on “Effect of Selenium and Lycopene on radiation sensitivity in Prostate Cancer patients relative to controls”. This study aims to elucidate the protective role of lycopene and selenium by food intake on prostate cancer.

Introduction provides the necessary information and background of the study.   The study population and methodology are described in full details and the statistical analysis is the appropriate one. Results are interesting and presented clearly and well.  Discussion is supported by the results and provides new insight on the matter.

Authors provide many perspectives for further study.

Response: We thank the Reviewer for the manuscript review and useful comments

Reviewer 3 Report

Overall comments:

The authors presented a manuscript in which the presence of prostate cancer biomarker were compared between sick and healthy groups, along with their lycopene/selenium plasma levels.

The article is written in good English, however, a complete grammar revision is needed, because there are many repeated terms and strangely worded phrases throughout the text. The scientific design of the study in very sound, the number of subjects is good, and the authors took care into taking major risk factors such as smoking into account. The lycopene and selenium antioxidant activity is well known, as well as, the inflammatory aspect of cancers in which the reducing effect makes sense. However, in the presented data I didn’t properly saw the discrimination of smoking vs. selenium/lycopene plasma levels. Given the nature of the study it may be difficult to acquire proper data from the patients, but to me it seems that the evaluation of the importance of the smoking factor and the lycopene/selenium would yield the best conclusions (In a logistic regression for example, it is possible to evaluate the variables importance, however, if you don’t have or it is too difficult to acquire the data at this point, please disregard this suggestion for this paper, and take this to your next work). Nevertheless, my main suggestions are for the authors to perform a complete revision of the grammar and to attend the evaluation of the smoking level & lycopene/selenium levels vs. PSA leves (variables importance in the predicting variable [prostate cancer marker]) (if is possible, if not disregard, for the article as is has its relevance, I am merely suggesting an upgrade).

Specific comments:

Simple summary: “Higher intake” is somewhat subjective, do the authors have a better estimation or a range?

For the keywords I suggest that the authors change to words not present in the title. Just a word of advice, but the search engine algorithms already take into account the words in the title, therefore, the keywords are opportunities for the authors to include the theme of the study and more subjective terms that a reader may type in the search bar, thus, increasing the visibility reach of your article

Materials and Methods

How the authors differentiated the High lycopene/selenium from the low lycopene/selenium diet groups? What was the threshold?

L162-163 “differentiate the differences” redundance

“Unpaired non-parametric Student’s t test was used to differentiate the differ- 162 ences between the two groups with regards to age, PSA levels, baseline and radia- 163 tion-induced micronuclei, nucleoplasmic bridges and nuclear buds in bi-nucleated cells, 164 we used the unpaired non-parametric Student’s t test.”

Repeated terms above

Author Response

Reviewer 3

Comments and Suggestions for Authors

Overall comments:

The authors presented a manuscript in which the presence of prostate cancer biomarker were compared between sick and healthy groups, along with their lycopene/selenium plasma levels.

  1. The article is written in good English, however, a complete grammar revision is needed, because there are many repeated terms and strangely worded phrases throughout the text.

Response: We performed a grammar revision and made the following amendments:

(i) restructured the sentence on lines 63-67 as indicated in our response to Reviewer 1.

(ii) amended the sentence on lines 75-80 as indicated in our response to Reviewer 1.

(iii) Deleted the sentence on lines 113-114.

(iv) Deleted the phrase "we used the unpaired non-parametric Student's t-test"  on line 165 to avoid its repetition within the same sentence..

The scientific design of the study in very sound, the number of subjects is good, and the authors took care into taking major risk factors such as smoking into account. The lycopene and selenium antioxidant activity is well known, as well as, the inflammatory aspect of cancers in which the reducing effect makes sense.

  1. However, in the presented data I didn’t properly saw the discrimination of smoking vs. selenium/lycopene plasma levels. Given the nature of the study it may be difficult to acquire proper data from the patients, but to me it seems that the evaluation of the importance of the smoking factor and the lycopene/selenium would yield the best conclusions (In a logistic regression for example, it is possible to evaluate the variables importance, however, if you don’t have or it is too difficult to acquire the data at this point, please disregard this suggestion for this paper, and take this to your next work).
  2. Nevertheless, my main suggestions are for the authors to perform a complete revision of the grammar and to attend the evaluation of the smoking level & lycopene/selenium levels vs. PSA leves (variables importance in the predicting variable [prostate cancer marker]) (if is possible, if not disregard, for the article as is has its relevance, I am merely suggesting an upgrade).

Response for points 2 &3: We performed a multiple regression analysis with respect to PSA and found that although Selenium (β= -0.187, p=0.009) and Lycopene β= -0.153, p=0.047)  are significantly associated with  PSA, this was not evident for smoking (β= -0.017, p = 0.767)

 Specific comments:

  1. Simple summary: “Higher intake” is somewhat subjective, do the authors have a better estimation or a range?

Response: We amended the term "Higher intake" in the simple summary to "Higher intake of foods that can raise plasma concentration to a level greater than 120umol for Selenium and 0.25ug/ml for Lycopene".

  1. For the keywords I suggest that the authors change to words not present in the title. Just a word of advice, but the search engine algorithms already take into account the words in the title, therefore, the keywords are opportunities for the authors to include the theme of the study and more subjective terms that a reader may type in the search bar, thus, increasing the visibility reach of your article

Response: We deleted the key words "selenium" and "lycopene" because they are already in the title and replaced them with the words "micronutrients" and "micronuclei"

  1. Materials and Methods

How the authors differentiated the High lycopene/selenium from the low lycopene/selenium diet groups? What was the threshold?

Response: Subjects were classified as high or low selenium intake subjects and/or high or low  lycopene intake subjects based on their plasma concentration of these micronutrients using threshold values of > 120umol for selenium and > 0.25ug/ml for lycopene.  This was explained in the Materials and Methods section (lines 165-171).

  1. L162-163 “differentiate the differences” redundance

“Unpaired non-parametric Student’s t test was used to differentiate the differences between the two groups with regards to age, PSA levels, baseline and radiation-induced micronuclei, nucleoplasmic bridges and nuclear buds in bi-nucleated cells, used the unpaired non-parametric Student’s t test.”

Repeated terms above

Response We deleted the redundant phrase in the relevant sentence which now reads as follows: 

"Unpaired non-parametric Student’s t test was used to determine the significance of differences between the two groups with regards to age, PSA levels, baseline and radiation-induced micronuclei, nucleoplasmic bridges and nuclear buds in bi-nucleated cells.

Reviewer 4 Report

It would be helpful for the authors to compare concentrations of  selenium and lycopene between the cases and controls (in Table 1) and enumerate the numbers of samples for each that had low and high concentrations of the nutrients (lycopene, selenium, lycopene+selenium)

For their comparisons of radiation-induced DNA damage, the authors make several comparisons between cases and controls by blood concentrations of selenium and lycopene (Figures 2 and 3); however, if the stated objective is to determine the impact of selenium/lycopene status on DNA damage markers, the more appropriate comparisons would be between high/low status within cases and within controls.  The current comparisons between case/control status within low and high lycopene/selenium level simply reflect differences between cases and controls, which reflects their difference in cancer status.  If these comparisons were made, they should be mentioned - even if insignificant.  These comparisons are made in Figure 4

For the comparisons of high selenium and high lycopene vs low selenium and low lycopene, were samples with high/low or low/high omitted? 

Under results, the headers state "Effect of low....etc.)  These are cross-sectional analyses and would be better stated as "associations". 

Did the authors consider any analyses excluding current smokers - given the differences in distribution of smoking between cases and controls and the possible impact of smoking on selenium/lycopene status.

Conclusions:  the conclusion statement suggests that prostate cancer patients are more prone to DNA damage...because they are deficient in selenium and lycopene.  The authors should consider restating this, as their data show prostate cancer patients that have high selenium and/or lycopene status.

Author Response

Reviewer 4:

It would be helpful for the authors to compare concentrations of selenium and lycopene between the cases and controls (in Table 1) and enumerate the numbers of samples for each that had low and high concentrations of the nutrients (lycopene, selenium, lycopene+selenium)

Response: As suggested we have included the concentrations of selenium and lycopene for cases and control in Table 1. Furthermore, the details of sample numbers are included in figure legend description for Figures 2, 3 and 4.

For their comparisons of radiation-induced DNA damage, the authors make several comparisons between cases and controls by blood concentrations of selenium and lycopene (Figures 2 and 3); however, if the stated objective is to determine the impact of selenium/lycopene status on DNA damage markers, the more appropriate comparisons would be between high/low status within cases and within controls.  The current comparisons between case/control status within low and high lycopene/selenium level simply reflect differences between cases and controls, which reflects their difference in cancer status.  If these comparisons were made, they should be mentioned - even if insignificant.  These comparisons are made in Figure 4

Response: We have made these comparisons and reported – values for only those that were significantly different in Figure 2, 3, 4, respectively. For clarity we have included a statement as follows: “P-values are only indicated for those comparisons that are significant”

For the comparisons of high selenium and high lycopene vs low selenium and low lycopene, were samples with high/low or low/high omitted? 

Response: Samples with high/low or low/high selenium and lycopene are now shown as a supplementary figure; they  were not significantly different from each other. Further details are included in result section (Section 3.4)

Under results, the headers state "Effect of low....etc.)  These are cross-sectional analyses and would be better stated as "associations". 

 Response: Agreed and we have made the necessary correction

Did the authors consider any analyses excluding current smokers - given the differences in distribution of smoking between cases and controls and the possible impact of smoking on selenium/lycopene status.

Response: We did not consider any analysis excluding current smokers because the regression analysis showed no effect of smoking on micronutrient status (β= -0.017, p = 0.767).

Conclusions:  the conclusion statement suggests that prostate cancer patients are more prone to DNA damage...because they are deficient in selenium and lycopene.  The authors should consider restating this, as their data show prostate cancer patients that have high selenium and/or lycopene status.

Response: We amended the first sentence in the Conclusion section to read as follows:

“In conclusion, the results of our study provide important evidence that prostate cancer patients who are deficient in selenium and/or lycopene may be more prone to DNA damage induced by ionising radiation.”

Round 2

Reviewer 3 Report

All suggestions/comments were answered properly.

Reviewer 4 Report

Thank you for addressing my comments.